# Genetic Variant *ABCC1* rs45511401 Is Associated with Increased Response to Statins in Patients with Familial Hypercholesterolemia

**DOI:** 10.3390/pharmaceutics14050944

**Published:** 2022-04-27

**Authors:** Carolina Dagli-Hernandez, Jéssica Bassani Borges, Elisangela da Silva Rodrigues Marçal, Renata Caroline Costa de Freitas, Augusto Akira Mori, Rodrigo Marques Gonçalves, Andre Arpad Faludi, Victor Fernandes de Oliveira, Glaucio Monteiro Ferreira, Gisele Medeiros Bastos, Yitian Zhou, Volker M. Lauschke, Alvaro Cerda, Mario Hiroyuki Hirata, Rosario Dominguez Crespo Hirata

**Affiliations:** 1Department of Clinical and Toxicological Analyses, School of Pharmaceutical Sciences, University of Sao Paulo, Sao Paulo 05508-000, Brazil; jessica.bassani.borges@gmail.com (J.B.B.); elisangela.idpc@gmail.com (E.d.S.R.M.); recarolinecf@gmail.com (R.C.C.d.F.); augusto.akira.mori@usp.br (A.A.M.); fo.victor@hotmail.com (V.F.d.O.); glauciom.ferreira@gmail.com (G.M.F.); mhhirata@usp.br (M.H.H.); 2Department of Physiology and Pharmacology, Karolinska Institutet, 171177 Stockholm, Sweden; yitian.zhou@ki.se (Y.Z.); volker.lauschke@ki.se (V.M.L.); 3Laboratory of Molecular Research in Cardiology, Institute Dante Pazzanese of Cardiology, Sao Paulo 04012-909, Brazil; gimebastos@gmail.com; 4Medical Clinic Division, Institute Dante Pazzanese of Cardiology, Sao Paulo 04012-909, Brazil; rodrigomg@gmail.com (R.M.G.); aaf.2312@gmail.com (A.A.F.); 5Department of Teaching and Research, Real e Benemerita Associação Portuguesa de Beneficiencia, Sao Paulo 01323-001, Brazil; 6Dr. Margarete Fischer-Bosch Institute of Clinical Pharmacology, Auerbachstr. 112, 70376 Stuttgart, Germany; 7University of Tuebingen, Geschwister-Scholl-Platz, 72074 Tübingen, Germany; 8Center of Excellence in Translational Medicine, CEMT-BIOREN & Department of Basic Sciences, Universidad de La Frontera, Av. Alemania 0458, Temuco 4810296, Chile; alvaro.cerda@ufrontera.cl

**Keywords:** statin, lipid response, familial hypercholesterolemia, pharmacogenetics, adverse drug events, myalgia, lipid-lowering drugs

## Abstract

Statins are the first-line treatment for familial hypercholesterolemia (FH), but response is highly variable due to genetic and nongenetic factors. Here, we explored the association between response and genetic variability in 114 Brazilian adult FH patients. Specifically, a panel of 84 genes was analyzed by exon-targeted gene sequencing (ETGS), and the functional impact of variants in pharmacokinetic (PK) genes was assessed using an array of functionality prediction methods. Low-density lipoprotein cholesterol (LDL-c) response to statins (reduction ≥ 50%) and statin-related adverse event (SRAE) risk were assessed in carriers of deleterious variants in PK-related genes using multivariate linear regression analyses. Fifty-eight (50.8%) FH patients responded to statins, and 24 (21.0%) had SRAE. Results of the multivariate regression analysis revealed that *ABCC1* rs45511401 significantly increased LDL-c reduction after statin treatment (*p* < 0.05). In silico analysis of the amino-acid change using molecular docking showed that *ABCC1* rs45511401 possibly impairs statin efflux. Deleterious variants in PK genes were not associated with an increased risk of SRAE. In conclusion, the deleterious variant *ABCC1* rs45511401 enhanced LDL-c response in Brazilian FH patients. As such, this variant might be a promising candidate for the individualization of statin therapy.

## 1. Introduction

Familial hypercholesterolemia (FH) is a primary dyslipidemia with frequent monogenic inheritance and autosomal dominant transmission [1]. Globally, FH prevalence was estimated at 1:313 in the heterozygous and 1:400,000 in the homozygous form, which implies that in total more than 30 million individuals are affected worldwide [2]. One of the main characteristics of FH is elevated plasma concentrations of low-density lipoprotein cholesterol (LDL-c) and early coronary artery disease (CAD) [3]. FH results from functional mutations in *LDLR*, *APOB*, and *PCSK9*, which encode proteins that regulate cholesterol homeostasis [3,4].

Statins are highly effective in reducing plasma LDL-c and are the first-line treatment for FH patients. Statins act via inhibition of 3-hydroxi-3-methylglutaryl Coenzyme A reductase (HMGR), mainly in hepatocytes, decreasing cholesterol de novo biosynthesis [5]. However, there is a great inter-individual variability of response, with an estimate of only approximately 20% of FH patients achieving therapeutic goals [6]. Statin-related adverse events (SRAE) have also been widely studied, and statin-associated muscular events (SAMS) in particular are commonly observed with frequencies ranging from 7 to 29% [7,8].

Many nongenetic factors, including gender, age, smoking status, diabetes, ethnicity, and exercise, have been reported as predictors of statin response and SRAE [9,10]. In addition to nongenetic factors, genetic variants have been shown to affect statin response and the risk of SRAE [10,11].

*SLCO1B1* is the most studied gene involved in statin pharmacokinetics. There is solid evidence that rs4149056 (c.521T>C), which is part of the *SLCO1B1*5*, *SLCO1B1*15*, and *SLCO1B1*17* alleles, is a decreased-function variant that predisposes to simvastatin-induced myopathy [12,13]. *SLCO1B1*5* increases statin plasma levels and the risk of myalgia, reaching a frequency of 50% in homozygous individuals against 19% in noncarriers [14].

Some studies focused on understanding the impact of genetic variants in LDL-c reduction. In addition to *SLCO1B1*, other drug transporters, such as *ABCB1*, have been widely studied in this context. *SLCO1B1* rs2306283 (c.388A>G), for example, was associated with a more pronounced reduction in LDL-c after treatment with atorvastatin and may be a predictor of therapeutic response [15]. We also recently found that *SLCO1B1*15* and variants in *SLCO1B3* and *ABCB11* delayed rosuvastatin response in an FH patient, without jeopardizing LDL-c reduction after 12 weeks of treatment [16]. Furthermore, variants in drug-metabolizing enzymes, such as *CYP3A4*22*, have been associated with higher LDL-c reduction [17], whereas our group reported that *CYP3A5*3* (rs776746) was associated with a lower reduction in total cholesterol, LDL-c, and HDL cholesterol (HDL-c) [18].

Statin pharmacogenetics studies have been performed in Brazilian cohorts and have brought important contributions, as we discussed in a recent review [19]. However, most Brazilian studies focused on nonfamiliar forms of hypercholesterolemia, and some of the results were not in agreement with reports from the literature. For example, in Brazilian patients, no association between SAMS and rs4149056 was found, whereas this variant constitutes the major risk factors in European individuals [20].

It is, thus, important to investigate statin pharmacogenetics in Brazilian FH patients as the prediction of treatment response is crucial for these high-risk individuals. Furthermore, those patients are exposed to higher statin doses and, therefore, are more susceptible to SRAE. In order to fill this gap, we studied the influence of pharmacogenetic variants detected by exon-targeted gene sequencing in FH patients.

## 2. Materials and Methods

### 2.1. Study Design and Patients

This study is a part of the FHBGEP project that aims to investigate genomic, epigenomic, and pharmacogenomic factors associated with FH in the Brazilian population [21]. Two hundred unrelated adult FH patients were recruited at three Brazilian Medical Centers from October 2014 to January 2020. FH was clinically diagnosed as possible (3–5 points), probable (6–8 points), or definite (>8 points) according to Dutch Lipid Clinic Network (DLCN) modified criteria [3,22].

Patients with the following comorbidities were excluded: liver failure, severe chronic kidney disease (estimated glomerular filtration rate, GFR < 30 mL/min/1.73 m^2^) and/or nephrotic syndrome, clinically uncontrolled neoplasms, positive serology for human immunodeficiency virus (HIV), hypothyroidism, and/or Cushing’s syndrome. Patients who withdrew from the study, were aged less than 18 years old, had no medical records available, or had no history of statin treatment were also excluded from the pharmacogenetics analysis.

The study protocol was approved by the Ethics Committees of the Institute Dante Pazzanese of Cardiology (CAAE #24618713.0.1001.5462, #24618713.0.1001.5462 and #05234918.4.0000.5462), and the School of Pharmaceutical Sciences (CAAE #24618713.0.3001.0067) of the University of Sao Paulo, and the Federal University of Rio Grande do Norte (CAAE #24618713.0.2001.5292), Brazil. The study was conducted according to good clinical practices and the Declaration of Helsinki guidelines (as revised in 2013). All subjects signed an approved written informed consent before enrollment.

### 2.2. Blood Samples and Laboratory Testing

Blood samples were obtained from fastened patients (at least 8 h) for DNA sequencing and laboratory testing: serum lipid profile (total cholesterol and fractions, triglycerides, apolipoproteins AI and B); glycemic profile (glucose and glycated hemoglobin); thyroid-related hormones (thyroid-stimulating hormone and thyroxine); liver function (aminotransferases) and muscle lesion (creatine kinase, CK); creatinine.

Plasma glucose, triglycerides, total cholesterol, and high-density lipoprotein cholesterol (HDL-c) were determined by colorimetric enzymatic methods. LDL-c and very-low-density lipoprotein cholesterol (VLDL-c) levels were calculated using Friedwald’s formula [23]. Creatinine, CK, alanine aminotransferase (ALT), and aspartate aminotransferase (AST) were determined by kinetic methods. Apolipoprotein (apo) AI and apo B were determined by immunoturbidimetry. Glycated hemoglobin (HbA1c) was determined by high-performance liquid chromatography (HPLC). These determinations described above were carried out using a Dimension RXL automatic analyzer (Siemens, Munich, Germany) following the manufacturer’s instructions.

Thyroid-stimulating hormone (TSH) and thyroxine (T4) were determined by sandwich-type enzymatic immunoassays, with detection by electrochemiluminescence, using a CENTAURO automatic analyzer (Siemens, Munich, Germany).

Laboratory external quality control was performed by the program of quality control of the Brazilian Society of Clinical Pathology.

### 2.3. Exon-Targeted Gene Sequencing

Genetic analyses were performed as previously described [21]. Briefly, genomic DNA was extracted from whole blood samples using QIAamp^®^ DNA Blood Maxi Kit (QIAGEN, Hilden, Germany). DNA quantification, purity (A260/A280 ratio), and integrity were analyzed using the QUBIT^®^ 2.0 fluorometer (Life Technologies, Forest City, IA, USA), NanoDrop^®^ ND-1000 spectrophotometer (Thermo Fisher Scientific Inc., Waltham, MA, USA), and 2200 TapeStation^®^ system (Agilent Technologies, Santa Clara, CA, USA).

FH- and pharmacogenetics-related genes were analyzed from a panel with 84 genes using an exon-targeted gene sequencing strategy [21]. Briefly, exons and regulatory regions were selected using Illumina’s Design Studio tools (https://accounts.illumina.com/, accessed on 20 January 2021). Good-quality genomic DNA was used for library construction using the Nextera Rapid Capture Custom Enrichment Kit (Illumina, San Diego, CA, USA). Clustering and paired-end sequencing reactions were performed using MiSeq^®^ Reagent kit V2 (300-cycles) in the MiSeq^®^ System (Illumina, San Diego, CA, USA). PhiX (1%) were used as library clustering and diversity controls. Sequencing data were analyzed using a variant discovery pipeline previously described [21].

The molecular diagnosis of FH was carried out by identifying variants previously associated with FH, such as gain-of-function variants in *PCSK9* or variants classified as pathogenic according to the American College of Medical Genetics and Genomics (ACMG) guidelines [24].

### 2.4. Clinical and Pharmacotherapeutic Data

Clinical and biodemographic data, including patient medical history, lifestyle information, medications in use, and adverse events, were obtained using a questionnaire and clinical examination, as previously described [21].

Information on pharmacotherapy and laboratory tests was also obtained from medical records. To mitigate information bias, the time between the medical visit and the corresponding laboratory test was set to a maximum of 30 days. Baseline LDL-c was considered the highest plasma level without statin treatment for at least 30 days when clearly indicated in the medical record. On-treatment LDL-c was defined as the lowest level with statin treatment.

Patients were considered responders if they reached an LDL-c reduction of at least 50% and nonresponders if they did not reach the therapy target [3,25]. Absolute LDL-c target was set according to the CAD risk stratification defined by the Update of the Brazilian Guideline for FH [3]: (i) very high risk: patients carrying manifested CAD (history of AMI, angina *pectoris*, previous myocardial revascularization, or ischemic or transitory cerebrovascular event); (ii) high risk: primary prevention with baseline LDL-c > 400 mg/dL, baseline LDL-c > 310 mg/dL with one high-risk factor (tobacco smoking, male gender, or HDL-c < 40 mg/dL), or baseline LDL-c > 190 mg/dL with two high-risk factors; (iii) intermediate risk: primary prevention without high-risk factors.

The therapy target for each risk group was the following: (i) very high risk: LDL-c reduction ≥ 50% and on-treatment LDL-c < 50 mg/dL; (ii) HIGH risk: LDL-c reduction ≥ 50% and on-treatment LDL-c < 70 mg/dL; (iii) intermediate risk: LDL-c reduction ≥ 50% and on-treatment LDL-c < 70 mg/dL.

FH patients were grouped according to the type and intensity of the statin therapy and the clinical response. Treatment intensity was established according to the American College of Cardiology/American Heart Association and the Brazilian guideline criteria, with moderate intensity (simvastatin 20–40 mg, atorvastatin 10–20 mg, or rosuvastatin 5–10 mg) or high intensity (simvastatin 80 mg + ezetimibe 10 mg, atorvastatin 40–80 mg or rosuvastatin 20–40 mg) [3,26]. Drug–drug interactions were annotated when a concomitant medication could inhibit or induce enzymatic activity and affect statin response [27]. SRAE were considered when clearly stated by the cardiologist as associated with statin therapy and were followed by dose reduction or change of statin [3]. Reduced adherence was considered for patients who reported at least one event of nonadherence to statin or ezetimibe [28].

### 2.5. Pharmacogenetic Analyses

A total of 23 genes involved in pharmacokinetics (PK) of statins, including cytochrome P450 (CYP) and uridine 5′-diphospho-glucuronosyltransferase (UGT) enzymes, as well as ATP-binding cassette (ABC) and solute carrier (SLC) transporters, were analyzed (Appendix A).

An optimized prediction model was used to evaluate the functional impact of variants in PK-related genes [29]. Briefly, missense, stop-gain, and stop-loss variants were analyzed using ANNOVAR [30] to assess the pathogenicity scores of five algorithms (LRT, Mutation Assessor, PROVEAN, VEST3, and CADD). Next, the PK-optimized prediction model was used, and variants were classified according to the functionality prediction score (FPS) as neutral (FPS < 0.5), deleterious (FPS > 0.5), or loss-of-function (LOF) (FPS = 1.0). Splicing site and frameshift variants were considered deleterious when they were classified as pathogenic or with decreased or increased activity in ClinVar (https://www.ncbi.nlm.nih.gov/clinvar/, accessed on 13 September 2021) and PharmVar (https://www.pharmvar.org/, accessed on 25 September 2021). Moreover, the functional impact of variants in splice sites was performed using ANNOVAR’s dbNSFP v4.2 in silico algorithm (accessed on 25 September 2021), followed by manual checking for the proximity to the branch point. Frameshift variants were considered deleterious. Variants were defined as novel if they were not reported in the dbSNP database (dbSNP build 155).

### 2.6. Molecular Modeling

The impact of deleterious genetic variants on the interaction between the protein and the statin ligands (simvastatin, atorvastatin, and rosuvastatin) was assessed using molecular modeling analysis as previously described [21].

Briefly, amino-acid sequences of reference proteins were downloaded from the Uniprot database (https://www.uniprot.org/help/uniprotkb, accessed on 10 October 2021), and three-dimensional models were generated using AlphaFold2 pipeline (https://github.com/deepmind/alphafold, accessed on 10 October 2021). Protein models (reference and variants) were prepared by adding hydrogen atoms, fixing missing side chains, removing sulfate ions and other crystallization buffer molecules such as glycerol, and minimizing by Biopolymer in Sybyl X suite (https://www.certara.com/, accessed on 10 October 2021). The ligands (simvastatin, atorvastatin, and rosuvastatin) were built using Spartan’14 (Wavefunction, Inc., Irvine, CA, USA). The minimization of ligands was performed using MMFF94 molecular mechanics method of Spartan’14 package.

Molecular docking was performed by GOLD 2020.3 (CCDC, Cambridge, UK) software. The docking runs were carried out with default settings and coordinates grid (10 Å) directed to Gly671 (MRP1). The best-ranked docking poses of statins were determined accordingly to the GoldScore fitness function and visual inspection of poses

### 2.7. Statistical Analyses

Statistical analyses were performed using RStudio V 4.0.3 (RStudio, Inc., Boston, MA, USA) and GraphPad Prism V8 (Sigma, San Diego, CA, USA). A cutoff *p*-value < 0.05 was used for statistical significance.

The distribution of the continuous variables was evaluated by the Kolmogorov–Smirnov test, and those with normal distribution are shown as the mean and SD and were compared using *t*-test. Continuous variables with skewed distribution are shown as the median and interquartile range and were compared using the Mann–Whitney test. For comparisons of continuous variables, Benjamini–Hochberg correction was used to adjust *p*-values, considering a false discovery rate (FDR) of 10%. Categorical variables were compared by chi-square or Fisher’s exact tests.

SNPassoc R package version 2.7 was used to analyze genotype and allele frequencies of the variants and Hardy–Weinberg equilibrium (HWE). Genetics package version 1.3.8.1.3 was used to calculate linkage disequilibrium.

Univariate and multivariate linear and logistic regression analyses were performed to investigate the influence of deleterious genetic variants on statin response and SRAE in FH patients. In univariate regression analyses, *p*-values were corrected using Benjamini–Hochberg correction for multiple tests. In multivariate regression analyses models, BMI, baseline LDL-c, treatment intensity, ezetimibe use, and SRAE (for analysis of statin response only) were used as covariates.

## 3. Results

### 3.1. Characteristics of the Individuals and Molecular Diagnosis

Of the 200 FH patients selected for this study, 86 were excluded due to lack of information from medical records: 19 did not use lipid-lowering medication; 55 did not have baseline laboratory data; six did not have on-treatment data; six did not have a medical record available.

Biodemographic and clinical characteristics of 114 FH patients are described in Table 1. Most patients were white (53.5%), female (71.9%), and clinically diagnosed FH as defined (41.2%), probable (27.2%), and possible (31.6%) according to the modified DCLN criteria. Most patients were at very high risk (56.1%) and high risk (9.7%) of CAD. The molecular diagnosis was confirmed for 35 (30.7%) patients, who carried pathogenic or likely pathogenic variants in *LDLR* (32) and *APOB* (1), according to ACMG classification, and a GOF variant in *PCSK9* (2) previously associated with FH. No pathogenic or likely pathogenic variants were found in *LDLRAP1* in this cohort (Table 2).

### 3.2. Statin Response

#### 3.2.1. Therapy Targets

A total of 58 (50.8%) FH patients were considered responders, and 56 (49.2%) were considered nonresponders to statin treatment. Clinical and molecular diagnosis of FH variables had similar results between responders and nonresponders, with the exception of median BMI and frequency of obesity and alcohol consumption, which were higher in nonresponders (*p* < 0.05) (Table 1). No difference was observed in FH clinical diagnosis between nonresponders and responders. Most patients were of very high risk (56.1%), intermediate risk (34.2%), and high risk (9.7%). The risk was similarly distributed in responders and nonresponders.

Most patients were treated with atorvastatin (79.8%), followed by simvastatin (10.5%) and rosuvastatin (9.6%). The type and intensity of statin therapy did not differ between responders and nonresponders (*p* > 0.05), but there was a significant association with ezetimibe, which was more prescribed in responders (*p* = 0.046). Regarding drug interactions, a total of 10 (8.8%) patients were taking amlodipine, an inhibitor of CYP3A4, but no difference was observed between responders and nonresponders (*p* > 0.05). One patient was also taking carbamazepine, which is an inducer of CYP3A4. Reduced adherence to therapy was similar between responders and nonresponders (*p* > 0.05). SAMS and other SRAE were more frequent in responders than nonresponders (*p* = 0.001).

When considering the absolute therapy target, 100 FH patients (87.7%) did not achieve optimum LDL-c levels after therapy. None of the patients of the CAD very-high-risk group reached an on-treatment LDL-c < 50 mg/dL. Furthermore, only two patients (18.2%) of the CAD high-risk group reached an LDL-c < 70 mg/dL, and 12 (30.8%) of the CAD intermediate risk group reached an LDL-c < 100 mg/dL (Appendix A).

Baseline and post-treatment values of serum lipid profile in responders and nonresponders are shown in the Figure 1. Responders had higher baseline total cholesterol and LDL-c than nonresponders, as well as lower on-treatment concentrations (*p* < 0.05) (Appendix A). As expected, responders had a higher reduction in total cholesterol (absolute and percentage change), LDL-c (absolute and percentage change), and triglycerides (percentage change) after treatment than nonresponders (*p* < 0.05) (Figure 1 and Appendix A). ApoAI, apoB, glucose, and insulin concentrations on treatment were higher in nonresponders than in responders (*p* < 0.05), whereas other variables were not significantly different between the groups (Appendix A).

As expected, individuals on high-intensity treatment showed lower post-treatment total cholesterol (*p* = 0.011) and triglycerides (*p* = 0.004) than individuals on moderate-intensity treatment, but no difference was observed in other lipid parameters (Appendix A). Furthermore, upon high-intensity treatment, the reductions (percentage change) in total cholesterol, LDL-c and triglycerides, and HDL-c increase were markedly higher. Patients taking ezetimibe in combination with statins had higher baseline total cholesterol and LDL-c concentrations (*p* < 0.05) and higher total cholesterol and LDL-c reductions (*p* < 0.05) than nonusers (Appendix A).

#### 3.2.2. Statin-Related Adverse Events

A total of 24 (21.0%) patients experienced SRAE, which included SAMS (19 patients, 79.1%), stomach pain (four patients, 16.7%), diarrhea (one patient, 4.2%), urinary tract infection (one patient, 4.2%), increased hepatic enzymes (one patient, 4.2%), and joint pain (one patient, 4.2%). Biodemographic characteristics of these patients are shown in Appendix A. The SRAE group had a higher frequency of xanthomas, FH-related pathogenic variants, a higher frequency of pathogenic variants in *LDLR*, and reduced adherence to statins and ezetimibe (*p* < 0.05). Interestingly, the prevalence of nonresponders was lower in SRAE group compared to no SRAE (*p* = 0.001).

Differences were also observed in the lipid profile of FH patients who experienced or did not experience SRAE (Appendix A). The SRAE group showed higher total cholesterol and LDL-c reductions compared to the no SRAE group (*p* < 0.05). Baseline total cholesterol and LDL-c were also higher in SRAE group, while the on-treatment total cholesterol was lower (Appendix A).

### 3.3. Variants in PK-Related Genes

Targeted sequencing identified 355 variants across 23 PK-related genes: 169 missense, two stop-gain, one stop-loss, seven frameshift indel, three in-frame deletions, 15 in splicing region, 26 in the 5′UTR region, and 132 in the 3′UTR region (Appendix A). Of the total variants identified, 41 (11.5%) were novel. Data on these novel variants were submitted to NCBI (https://www.ncbi.nlm.nih.gov/sra/PRJNA662090, submitted on 8 September 2020).

The functional impact of variants (functional prediction score; FPS) was estimated using the ADME Prediction Framework (APF). For missense and stop-loss variants in PK genes, 61 variants with MAF > 1% were predicted to reduce function by >50% (Table 3). The most frequent variants were *SLCO1B3* rs60140950 (c.767G>C, LOF; MAF: 14.7%); *SLCO1B1* rs4149056 (c.521T>C; MAF: 11.0%), *CYP2C9*2* rs1799853 (c.430C>T, LOF; MAF: 8.8%), *CYP2D6* rs1065852 (c.941G>A, LOF; MAF: 6.0%), and *ABCC3* rs11568591 (c.3890G>A; MAF: 6.5%). Furthermore, we detected five novel deleterious variants (Appendix A).

A total of 16 splice-site variants in PK genes were considered deleterious according to the functional prediction algorithm (located at splice donor or splice acceptor regions) (Table 4). Two known deleterious splice variants, *CYP3A5*3* and *CYP3A5*6*, were detected in FH patients. *CYP3A5*3* (MAF: 49.6%) and *CYP2C8* rs2071426 (c.1275_1276del) (MAF: 45.7%) were the most frequent variants.

According to in silico functional analysis of frameshift and in-frame variants in PK-related genes, three in-frame variants were considered as likely deleterious, and seven frameshift variants were considered deleterious, including the novel variant *ABCC1* c.66del (Table 4).

### 3.4. Association Study between Variants in PK Genes and Statin Response

#### 3.4.1. LDL-c Reduction

To assess the influence of variants in PK genes on statin response, 24 deleterious variants detected in at least three carriers were analyzed. FH patients carrying the homozygous form of the minor allele were grouped with the heterozygous carriers and compared with noncarriers (dominant inheritance model). Figure 2 and Appendix A show the results for deleterious variants in PK genes with MAF > 5%.

Carriers of the deleterious variant *ABCC1* rs45511401 (c.2012G>T) T allele had greater on-treatment LDL-c reduction with either all statins or atorvastatin treatment (*p* < 0.001, adjusted *p* < 0.10). One patient was considered an outlier after showing an increase in LDL-c levels after statin treatment. The *SLCO1B1* rs4146056 c.521C allele, a known deleterious variant, and *CYP3A5*3*, a nonfunctional splicing variant, were not associated with statin response (Appendix A).

Univariate linear regression analysis showed that *ABCC1* c.2012T allele contributed for an additional reduction of 18.8% in LDL-c after statin therapy (*p* = 0.016, adjusted *p* = 0.096) (Appendix A). Baseline LDL-c and therapy intensity also enhanced LDL-c reduction, whereas BMI had an opposite effect (*p*-adjusted < 0.05). Multivariate linear regression analysis of variants in PK-related genes with MAF > 1.0% was performed adjusting each model only with nongenetic covariates (body mass index, baseline LDL-c, therapy intensity, and presence of SRAE). This analysis showed no association between *ABCC1* c.2012G>T or other variants and enhanced LDL-c reduction (Appendix A).

Next, we performed a multivariate linear regression analysis by including all deleterious variants with MAF > 10% in the model and adjusting for nongenetic covariates using a dominant model (Table 5). In this model, *ABCC1* c.2012T allele enhanced LDL-c reduction by 13.8% after statin therapy (*p* = 0.046).

Univariate logistic regression analysis of with variants in PK-related genes and nongenetic variables showed that higher baseline LDL-c, ezetimibe use, manifestation of SRAE or myopathy, and lower BMI were associated with higher likelihood of being responder to statin (*p* < 0.05) (Appendix A). However, the association with ezetimibe use and BMI was not sustained after correction (*p* > 0.05).

Multivariate logistic regression analysis showed that variants in PK-related genes were not associated with the likelihood of being a responder to statins, even after adjustment with nongenetic covariates (Appendix A).

#### 3.4.2. Molecular Modeling Results

Molecular modeling analysis was performed to explore the influence of the missense variant *ABCC1* rs45511401 (c.2012G>T, p.Gly671Val in MRP1) on statin binding. MRP1 reference (Gly671) and variant (Val671) models were prepared by adding hydrogen atoms, fixing missing side chains, removing sulfate ions and other crystallization buffer molecules such as glycerol and minimizing by Biopolymer in Sybyl X suite.

As shown in Figure 3, the variant Val671 resulted in shorter distances of MRP1 interactions with atorvastatin (2.1 Å), rosuvastatin (1.1 Å), and simvastatin (1.7 Å) compared to the reference Gly671 (4.1 Å, 3.7 Å, and 4.3 Å, respectively). These results indicate that the amino-acid change from glycine to valine in position 671 enhances the interaction of MRP1 with statins, possibly reducing efflux across the basolateral membrane of the hepatocytes. In this way, the variant would increase retention of statins within the liver, increasing the LDL-c response.

#### 3.4.3. Statin-Related Adverse Events

The association of deleterious variants in PK genes with MAF > 1% and nongenetic variables with SRAE was also assessed by univariate logistic regression analysis. Higher baseline LDL-c increased the risk of SRAE (*p* < 0.05). Reduced adherence, drug interaction with CYP3A4 inhibitor, and FH-related variants were also predictors of SRAE, but these associations were not maintained after corrections (adjusted *p* ≥ 0.05) (Appendix A). Deleterious variants in PK genes were not associated with SRAE according to univariate logistic regression analysis (Appendix A) or multivariate logistic regression analysis after adjustment with nongenetic covariates (*p* = 0.067) (Table 6).

## 4. Discussion

In PK-related genes, most associations with statin response described in previous studies were observed with *ABCB1*, *SLCO1B1*, *CYP3A4*, *CYP2C9*, and *CYP3A5* variants. For example, *ABCB1* rs2032582 (c.2677T>G/A) and *ABCB1* rs1045642 (c.3435C>T) were associated with better statin response in some studies [31,32,33,34], as well as *SLCO1B1* rs2306283 (c.388A>G) [15], while *CYP3A5*3* was shown to be associated with lower statin response [18]. However, there is still controversy about these associations, and no variant showed strong evidence of influencing statin response.

In this study, carriers of *ABCC1* rs45511401 showed improved statin response. MRP1 (encoded by *ABCC1*) is an ABC membrane transporter highly expressed in the thymus, skeletal muscle tissue, kidney, urinary bladder, and gastrointestinal tract. It promotes the efflux of drugs, including statins and their metabolites, from hepatocytes to the bloodstream [19]. MRP1 is a highly conserved protein [35], but several variants, deleterious or not, have been identified worldwide [36,37].

Previous studies have reported the importance of *ABCC1* rs45511401 in pharmacogenetics. It has been associated with febrile neutropenia in breast cancer patients undergoing treatment with 5-fluorouracil, epirubicin, and cyclophosphamide chemotherapy [38]. Furthermore, in vitro recombinant overexpression of *ABCC1* c.2012T (p.671Val) retained approximately 20% more doxorubicin compared to the reference protein, indicating that the variant might reduce function, which could explain increased doxorubicin-associated acute cardiac toxicity [39].

When considering statin response, however, there are conflicting results on the influence of *ABCC1* rs45511401. In this study, this variant was associated with increased LDL-c reduction after treatment with all statins, supporting its reduced function. In contrast, a study with Iranian hypercholesterolemic patients showed that carriers of c.2012T allele had a lower percentage reduction in LDL-c and total cholesterol compared to GG carriers when on atorvastatin 10 mg/day treatment (*p* = 0.02), but no difference was observed in patients using atorvastatin 20 or 40 mg (*p* = 0.81) [40]. Similarly, a previous study from our group showed no association between this variant and LDL-c reduction in Brazilian hypercholesterolemic patients, but *ABCC1* mRNA levels were reduced in mononuclear cells of patients treated with atorvastatin 10 mg/day compared to baseline levels [33].

The prediction framework score used in this study indicated that *ABCC1* rs45511401 is potentially deleterious. This is corroborated by our molecular docking analyses, which showed a stronger interaction between the MRP1 variant Val671 and statins. Although both reference and variant amino acids are nonpolar, the in silico characterization study showed that this change shifted the free energy of MRP1 [41]. This is possibly due to the special properties of the reference amino acid, glycine. Glycine has a hydrogen in its side-chain, unlike other amino acids that carry a carbon chain. This confers unique flexibility to glycine, allowing it to be in tight regions of proteins that are not easily accessible to other amino acids [42]. The change to valine, which does not contain these properties, can cause conformational changes in MRP1, making this region more accessible to substrates [42]. This stronger protein–ligand interaction possibly leads to a less efficient statin efflux from hepatocytes by retaining the statins bound to position 671 in MRP1. Since MRP1 acts in statin efflux from the liver, a possible mechanism through which p.Gly671Val increased statin response would be that the lower function of this protein led to an increased intracellular statin concentration in hepatocytes. This, in turn, could enhance the inhibition of HMGR and, therefore, potentiate the cholesterol-lowering effect.

A similar mechanism was proposed in a case report of a female FH patient with late rosuvastatin response previously published [16]. The patient underwent a 6 week rosuvastatin wash-out period, after which rosuvastatin 20 mg was reintroduced. However, after 6 weeks of treatment, her lipid profile did not show any changes from baseline, which could only be observed after 12 weeks of rosuvastatin treatment. The patient was a carrier of the deleterious variants *SLCO1B1*15, SLCO1B3* rs4149117 and rs7311358, *ABCB11* rs2287622, and LOF variant *CYP3A5*3*. Possibly, the effect of the deleterious variants in the influx proteins OATP1B1 (*SLCO1B1*) and OATP1B3 (*SLCO1B3*) led to a slow uptake of rosuvastatin by hepatocytes, which led to a lower response in the first 6 weeks. However, the patient still responded to rosuvastatin treatment after 12 weeks. This could be due to an accumulation of rosuvastatin in hepatocytes resulting from the effect of the deleterious variant in the efflux protein BSEP (*ABCB11*).

It is noteworthy that the previous studies were performed with lower doses of atorvastatin (10 to 40 mg) [33,40], while the majority of patients in this cohort were on high atorvastatin doses (40 or 80 mg). Clearer effects of *ABCC1* rs45511401 may be observed in higher statin doses. However, experimental investigations and observational studies with larger samples are necessary to clarify these disparities. However, we believe that *ABCC1* rs45511401 is an interesting candidate for the individualization of statin therapy, as it could help predicting a better statin response in FH patients.

Interestingly, the influence of *ABCC1* rs45511401 on the percentage LDL-c change was significant in the multiple regression linear analysis when considering deleterious variants in PK genes with MAF > 10%. This result possibly shows that the influence of pharmacogenetic variants is not isolated but depends on the burden of deleterious variants carried by each patient. Therefore, similarly to the discussion presented in our case report [16], the effect of each variant on statin response could be potentialized or annulated by the interaction with other variants, consequently leading to the phenotype observed. The analysis of the burden of variants and their interaction could be an approach for pharmacogenetic interpretation of FH patients. Since we had a limited number of patients, we could not analyze the effect of variants with lower MAF; however, this approach could be used for future pharmacogenetic studies with higher sample sizes to understand how these variants interact with each other.

Other common variants on PK genes were not shown to influence statin response in this study. The remaining variants were not consistently shown to impact statin response in previous studies. *CYP3A5*3*, for example, has been shown to decrease total cholesterol, LDL-c, and HDL-c reduction after atorvastatin treatment in Brazilian hypercholesterolemic patients [18], but no differences were observed in another study with Chilean hypercholesterolemic patients [43]. Other variants, such as *SLCO1B1*5*, have been shown to increase statin blood levels in previous studies, but did not show to impact statin response, which is in agreement with the results in our study [19].

In this study, no deleterious PK-related variants were significantly associated with increased risk of SRAE. We previously discussed the lack of association between SRAE and *SLCO1B1*5* and **15*, a well-described variant, in the Brazilian population in a recent review [19]. This is probably due to low sample sizes, which impaired the statistical power of the analysis in previous studies with Brazilian patients [19]. Although SRAEs were very frequent in this study, the size of the SRAE group is still small, which makes the association study difficult. Therefore, it is necessary to increase the sample size in order to study the association between genetic variants and SRAE in FH patients.

This study had some limitations. Firstly, we had a low sample size, which impaired the association study of deleterious variants, especially those with lower frequency. Secondly, this was an observational, retrospective study, which is susceptible to some biases, such as information bias. However, we mitigated these biases by establishing a rigorous protocol of medical record review and data selection. Another limitation is that we used an exon-targeted strategy instead of whole-genome sequencing, which could be an interesting approach to discover new loci involved in statin response. However, genome-wide association studies have been previously performed [44,45], and most of the genes shown to be relevant to statin response participate in cholesterol homeostasis and statin pharmacokinetics. For statin-related myopathy, a recent meta-analysis of whole-genome sequencing showed that no variants were associated with this event [46,47]. Thus, this study focused on the main genes that could be related to statin response.

## 5. Conclusions

In PK-related genes, the deleterious variant *ABCC1* rs45511401 (c.2012G>T) is a major contributor to LDL-c response, enhancing LDL-c reduction after statin treatment in Brazilian FH patients. *ABCC1* c.2012G>T causes a stronger interaction between MRP1 and statins, impairing their efflux. Therefore, this variant could be a promising marker for the individualization of statin therapy.

Variants in PK-related genes are not associated with increased risk of SRAE in FH patients.

## Figures and Tables

**Figure 1 pharmaceutics-14-00944-f001:**
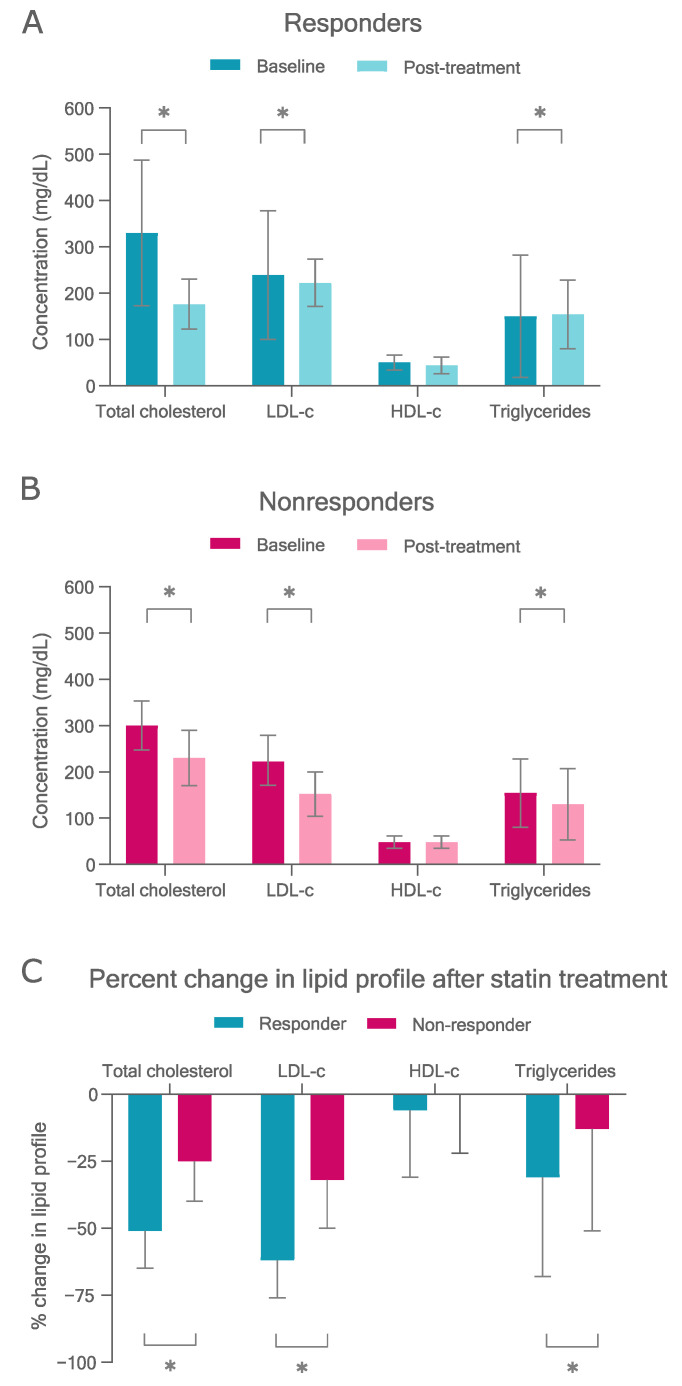
Plasma lipid profile in FH patients treated with lipid-lowering drugs. (**A**) Baseline and post-treatment mean values and standard deviation (SD) in responders (LDL-c reduction ≥ 50%). (**B**) Baseline and post-treatment mean values and SD in nonresponders (LDL-c reduction < 50%). (**C**) Plasma lipid response (mean values and SD of % change) in responder and nonresponder groups. * *p* < 0.05 (compared by *t*-test).

**Figure 2 pharmaceutics-14-00944-f002:**
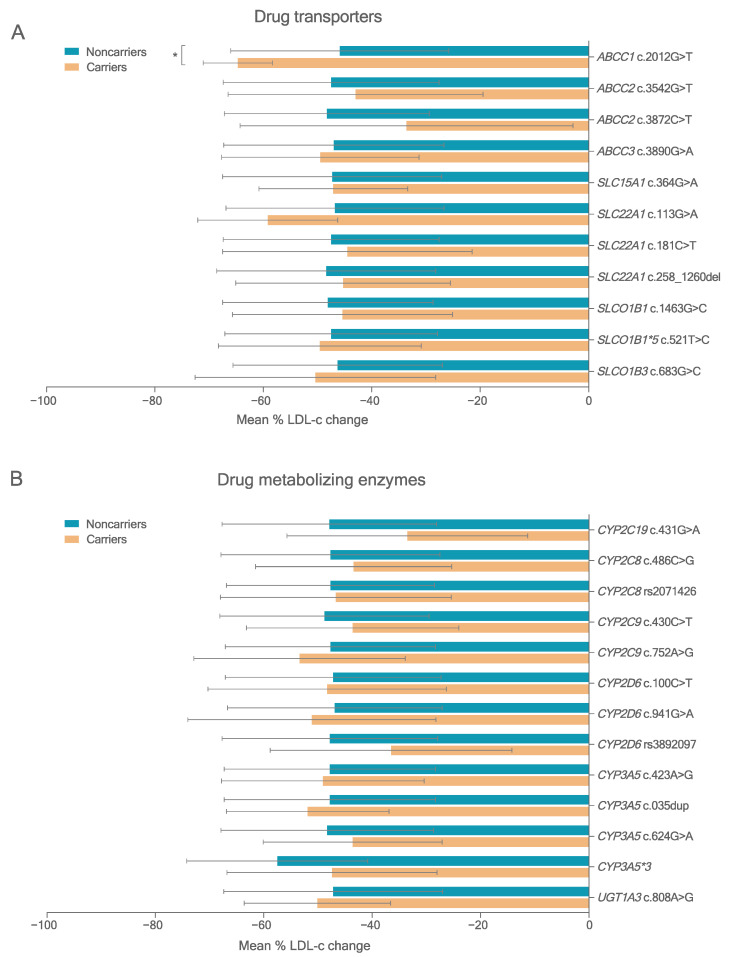
Mean LDL cholesterol response percentage change) after lipid-lowering treatment in FH patients carrying deleterious variants in PK genes (MAF > 5.0%). (**A**) Variants in ABC and SLC transporters. (**B**) Variants in CYP and UGT metabolizing enzymes. * *p* < 0.05 (compared by *t*-test).

**Figure 3 pharmaceutics-14-00944-f003:**
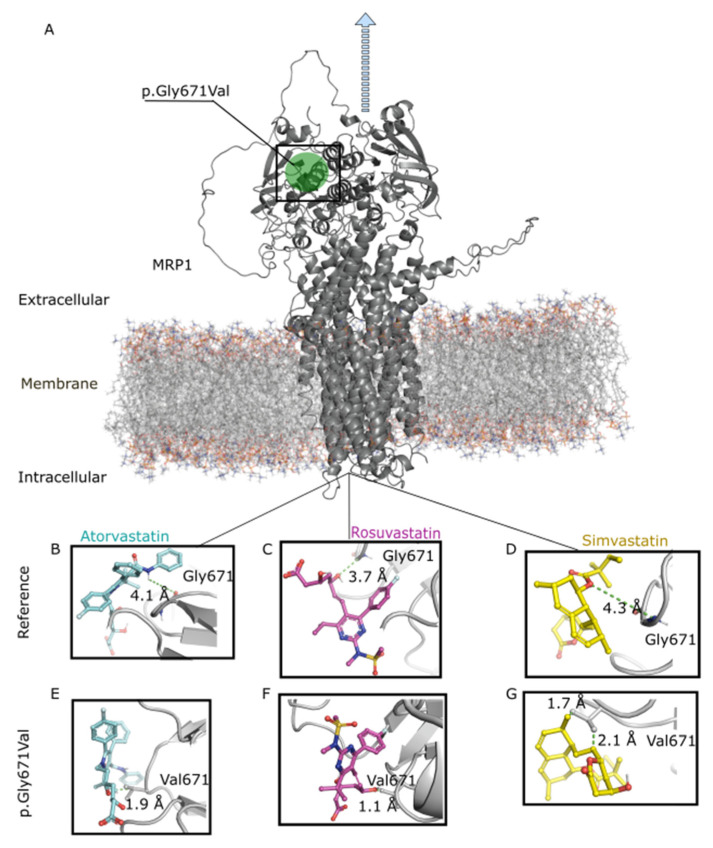
Molecular modeling analysis. Influence of *ABCC1* rs45511401 (c.2012G>T, p.Gly671Val in MRP1) on amino-acid interaction with statins. (**A**) Representation of MRP1 (encoded by *ABCC1*) anchored in the basolateral membrane of a hepatocyte. The blue arrow indicates the sense of statin efflux. (**B**–**D**). Interactions between reference MRP1 (Gly671) and atorvastatin, rosuvastatin, and simvastatin, respectively. (**E**–**G**). Interactions between MRP1 variant (Val671) and atorvastatin, rosuvastatin, and simvastatin, respectively.

**Table 1 pharmaceutics-14-00944-t001:** Biodemographic and clinical data of FH patients grouped according to statin response.

Variable ^a^		Total (114)	Responders (58)	Nonresponders (56)	*p*-Value
Age, years		57.1 (37.9–76.3)	54.9 (34.7–75.1)	57.6 (41.9–73.3)	0.261
Gender (female), %		71.9 (82)	69.0 (40)	75.0 (42)	0.611
Ethnicity, %	White	53.5 (54)	58.5 (31)	48.9 (23)	0.326
	Brown	31.7 (32)	24.5 (13)	38.3 (18)	
	Black	14.9 (15)	17.0 (9)	12.8 (6)	
Xanthomas, %		12.3 (14)	13.8 (8)	10.7 (6)	0.830
Arcus cornealis, %		17.9 (20)	14.0 (8)	21.8 (12)	0.407
FH clinical diagnosis ^b^, %	Defined or probable	68.4 (78)	75.9 (44)	60.7 (34)	0.124
	Possible	31.6 (36)	24.1 (14)	39.3 (22)	
FH molecular	FH variants	30.7 (35)	34.5 (20)	26.8 (15)	0.491
diagnosis, %	*APOB*	0.9 (1)	0.0 (0)	1.8 (1)	0.166
	*LDLR*	28.3 (32)	34.5 (20)	21.4 (12)	
	*PCSK9*	1.8 (2)	0.0 (0)	3.6 (2)	
	*LDLRAP1*	0.0 (0)	0.0 (0)	0.0 (0)	
Hypertension, %		62.5 (70)	60.3 (35)	64.8 (35)	0.770
Type 2 diabetes, %		21.6 (24)	26.3 (15)	16.7 (9)	0.316
Obesity, %		28.6 (32)	17.2 (10)	40.7 (22)	0.011
BMI, kg/cm^2^		27.7 (22.5–32.9)	26.3 (21.4–31.2)	28.2 (22.5–33.9)	0.011
Medical history, %	AMI	29.2 (33)	28.1 (16)	30.4 (17)	0.952
	CAD	40.0 (42)	44.0 (22)	36.4 (20)	0.550
	CVE	6.0 (6)	3.9 (2)	8.2 (4)	0.637
Alcohol consumption, %		25.0 (22)	14.6 (7)	37.5 (15)	0.007
Tobacco smoking, %		14.3 (16)	17.2 (10)	11.1 (6)	0.510
CAD risk, %	Very high risk	56.1 (64)	53.4 (31)	58.9 (33)	0.095
	High risk	9.7 (11)	15.5 (9)	3.6 (2)	
	Intermediate risk	34.2 (39)	31.0 (18)	37.5 (21)	
Lipid-lowering	Atorvastatin	79.8 (91)	77.6 (45)	82.1 (46)	0.275
treatment, %	Simvastatin	10.5 (12)	8.6 (5)	12.5 (7)	
	Rosuvastatin	9.6 (11)	13.8 (8)	5.4 (3)	
	Statins + Eze	36.8 (42)	46.6 (27)	26.8 (15)	0.046
Statin intensity, %	Moderate	14.0 (16)	6.9 (4)	21.4 (12)	0.050
	High	86.0 (98)	93.1 (54)	78.6 (44)	
Drug interactions, %	CYP3A4 inhibitors ^c^	10 (8.8)	7 (12.1)	3 (5.3)	0.349
	CYP3A4 inhibitors + inducers ^d^	1 (0.01)	0 (0.0)	1 (1.9)	-
Reduced adherence, %	Statins	15.9 (18)	17.2 (10)	14.5 (8)	0.893
	Ezetimibe	10.6 (12)	13.8 (8)	7.3 (4)	0.413
SRAE, %	SAMS	16.8 (19)	29.3 (17)	3.6 (2)	0.001
	Others ^e^	21.2 (24)	34.5 (20)	7.3 (4)	0.001

Number of patients in brackets. Patients with ≥50% LDL cholesterol reduction on statin treatment were classified as responders. Categorical variables were compared by chi-square test. Continuous variables are shown as median and interquartile range and were compared by Mann–Whitney test. AMI: acute myocardial infarction; BMI: body mass index; CAD: coronary artery disease; CVE: cerebrovascular event; Eze: ezetimibe; SRAE: statin-related adverse events ^a^ Data were not available for ethnicity (13 patients), *arcus cornealis* (two patients), hypertension (two patients), diabetes (three patients), BMI (four patients), obesity (two patients), history of AMI (one patient), CAD (nine patients), CVE (14 patients), tobacco smoking (two patients), alcohol consumption (26 patients), and age (two patients). ^b^ DCLN modified criteria. ^c^ All patients in this category used the CYP3A4 inhibitor amlodipine. ^d^ All patients in this category used the CYP3A4 inhibitor amlodipine and the CYP3A4 inducer carbamazepine. ^e^ Including also stomach pain (four patients), diarrhea (one patient), urinary tract infection (one patient), increased hepatic enzymes (one patient), and joint pain (one patient).

**Table 2 pharmaceutics-14-00944-t002:** FH-related pathogenic variants in FH patients (*n* = 114).

Gene	dbSNP Code	Variant	Amino-Acid Change	Type	In Silico Analysis ^a^	ACMG Classification	Number of Patients (Zygosity)
*APOB*	rs61744153	c.11477C>T	p.Thr3826Met	Missense	D	LD	1 (He)
*LDLR*	rs112029328	c.313+1G>A	-	Splice-site	NA	D	2 (He)
	rs121908026	c.530C>T	p.Ser177Leu	Missense	D	D	2 (He)
	rs875989902	c.533A>T	p.Asp178Val	Missense	D	LD	1 (He)
	rs121908039	c.551G>A	p.Cys184Tyr	Missense	D	D	1 (He)
	rs879254797	c.1118G>A	p.Gly373Asp	Missense	D	LD	2 (He)
	rs28942078	c.1285G>A	p.Val429Met	Missense	D	D	1 (He)
	rs28942079	c.1291G>A	p.Ala431Thr	Missense	D	D	1 (He)
	rs879254913	c.1463T>C	p.Ile488Thr	Missense	D	LD	2 (He)
	rs373646964	c.1474G>A	p.Asp492Asn	Missense	D	LD	1 (He)
	rs28941776	c.1646G>A	p.Gly549Asp	Missense	D	D	2 (He)
	rs137929307	c.1775G>A	p.Gly592Glu	Missense	D	LD	2 (He)
	rs753707206	c.1801G>C	p.Asp601His	Missense	D	LD	2 (He)
	rs879254687	c.818-2A>G	-	Splice-site	NA	D	1 (He)
	rs1135402774	c.1474del	p.Asp492fs	InDel	NA	D	1 (He)
	rs121908031	c.2043C>A	p.Cys681*	Stop-gain	D	D	6 (He)
	rs752596535	c.501C>G	p.Cys167*	Stop-gain	D	D	2 (He)
	rs1135402768	c.487C>T	p.Gln163*	Stop-gain	D	D	1 (He)
	rs875989887	c.-140C>A	-	5′UTR	NA	LD	1 (Ho)
	rs387906307	c.-138del-T	-	5′UTR	NA	LD	1 (He)
*PCSK9*	rs141502002	c.1405C>T	p.Arg469Trp	Missense	LN	Conflict ^b^	2 (He)

^a^ The functionality of missense, stop-gain, and stop-loss variants was assessed using the in silico prediction algorithms PolyPhen-2, Mutation Assessor, SIFT, PROVEAN, CADD, DANN, and FATHMM. ^b^ This variant is of unknown significance (VUS) according to the ACMG criteria, but it was reported as gain-of-function in previous studies. Therefore, it was considered pathogenic. ACMG: American College of Medical Genetics and Genomics; D: deleterious; He: heterozygous; Ho: homozygous; LD: likely deleterious; LN: likely neutral; NA: not applicable; UTR: untranslated region.

**Table 3 pharmaceutics-14-00944-t003:** Missense and stop-loss variants in PK-related genes (MAF > 1.0%) with deleterious functionality prediction score (FPS > 0.5).

Gene	Variant	NT Change	AA Change	Type	MAF (%)	MAF (gnomAD ^a^, %)	FPS
*CYP2C8*	rs1058930 (*CYP2C8*4*)	c.486C>G	p.Ile162Met	Missense	4.9	2.7	0.6
*CYP2C9*	rs1799853 (*CYP2C9*2)*	c.430C>T	p.Arg144Cys	Missense	8.8	6.8	1
	rs2256871 (*CYP2C9*9)*	c.752A>G	p.His251Arg	Missense	2.2	0.3	0.8
*CYP2C19*	rs17884712 (*CYP2C19*9*)	c.431G>A	p.Arg144His	Missense	2.2	0.1	0.8
*CYP2D6*	rs1065852 (*CYP2D6*10*)	c.100C>T	p.Pro34Ser	Missense	0.03 ^b^	12.3	1
	rs28371703	c.271C>A	p.Leu91Met	Missense	1.1	5.9	0.6
	rs1058172	c.941G>A	p.Arg314His	Missense	4.9	5.6	1
*CYP3A5*	rs6977165	c.423A>G	p.X141Trp	Stoploss	5.7	8.1	1
	rs10264272 (*CYP3A5*6*)	g.19787G>A	p.Lys208 =	Synonymous ^c^	3.1	0.7	1.0
*UGT1A3*	rs45449995	c.808A>G	p.Met270Val	Missense	2.2	1.6	0.75
*ABCC1*	rs45511401	c.2012G>T	p.Gly671Val	Missense	3.8	1.7	0.8
*ABCC2*	rs8187692	c.3542G>T	p.Arg1181Leu	Missense	2.7	0.6	0.8
	rs17216317	c.3872C>T	p.Pro1291Leu	Missense	3.3	0.2	0.8
*ABCC3*	rs11568591	c.3890G>A	p.Arg1297His	Missense	6.5	2.9	0.8
	rs141856639	c.3971G>A	p.Arg1324His	Missense	1.1	0.01	1
*SLC15A1*	rs8187820	c.364G>A	p.Val122Met	Missense	1.6	0.3	0.6
*SLC22A1*	rs2282143	c.1022C>T	p.Pro341Leu	Missense	1.1	4.4	0.8
	rs35888596	c.113G>A	p.Gly38Asp	Missense	2.2	0.4	1
	rs34059508	c.1393G>A	p.Gly465Arg	Missense	1.1	0.7	0.8
	rs12208357	c.181C>T	p.Arg61Cys	Missense	3.8	2.3	0.6
*SLCO1B1*	rs59502379	c.1463G>C	p.Gly488Ala	Missense	1.8	0.1	0.8
	rs4149056 (*SLCO1B1*5)*	c.521T>C	p.Val174Ala	Missense	11.0	11.2	0.8
*SLCO1B3*	rs60140950	c.767G>C	p.Gly228Ala	Missense	14.7	7.4	1

AA: amino acid; FPS: functionality prediction score; MAF: minor allele frequency; NT: nucleotide; PK: pharmacokinetics. ^a^ MAF obtained for Latino/Admixed Americans from gnomAD database v2.1.1 (https://gnomad.broadinstitute.org, accessed on 26 September 2021). ^b^ MAF obtained for *CYP2D6*10* variant when not in the presence of *CYP2D6*4* (linkage disequilibrium: *r*^2^ = 0.35). ^c^ Although this variant (*CYP3A5*6*) is synonymous, it has been described in the literature as low-function.

**Table 4 pharmaceutics-14-00944-t004:** In silico functional prediction of splice-site, frameshift, and in-frame variants in PK-related genes.

Gene	Variant	NT Change ^a^	Type	MAF (%)	MAF (gnomAD ^b^, %)	Prediction ^c^
*Splice-site variants*						
*ABCC1*	rs8187856	g.16146576C>G	Splice region	1.1	0.3	B
*ABCC2*	rs533334893	g.101552117G>A	Splice donor	0.5	0.0	D
*ABCC3*	rs11568607	g.48745787G>A	Splice region	2.2	0.6	B
*ABCG2*	rs34124189	g.89053790G>A	Splice region	0.5	0.1	B
*CYP1A2*	rs1288558234	g.75041241del	Splice region	0.5	0.1	B
	rs913188841	g.75041242C>G	Splice region	0.5	0.1	B
*CYP2C8*	rs11572078	g.96827126dup	Splice region	17.4	16.8	B
	rs2071426	g.5932A>G	Splice donor	23.9	15.4	D
*CYP2D6*	rs3892097 (*CYP2D6*4)*	g.6866G>A	Splice acceptor	2.2	11.1	D
*CYP3A5*	rs776746 (*CYP3A5*3*)	g.12083G>A	Splice acceptor	49.6	20.8	D
*SLC15A1*	rs8187827	g.99354731T>C	Splice region	0.5	1.4	B
*SLC22A1*	rs35854239	c.1275_1276del	Splice acceptor	45.7	NR	D
*SLCO1B1*	rs77271279	g.21329832G>T	Splice donor	0.9	0.2	D
*SLCO1B3*	rs3764009	g.21013948C>T	Splice region	16.3	79.0	B
	rs958332597	g.21032366C>T	Splice region	0.5	0.0	B
*Frameshift and in-frame variants*						
*ABCC1*	Novel	c.66del	Frameshift variant	0.5	NR	D
*CYP2D6*	rs5030656	c.88_690del	In-frame deletion	0.5	1.2	LD
		c.54del	Frameshift truncation	1.1	0.4	D
*CYP3A5*	rs200579169	c.2dup	Frameshift truncation	0.4	0.4	D
	rs41303343	c.1035dup	Frameshift variant	1.8	0.4	D
	rs547253411	c.372del	Frameshift truncation	0.4	0.03	D
*SLC22A1*	rs72552763	c.1258_1260del	Disruptive in-frame deletion	18.5	24.3	LD
*SLCO1B3*	rs780598056	c.333del	Frameshift truncation	0.5	0.0	D
	rs558592800	c.19_120insAATT	Frameshift elongation	0.5	0.01	D
*SLCO2B1*	rs60113013	c._14del	In-frame insertion	1.6	3.1	LD

B: benign; D: deleterious; LD: likely deleterious; MAF: minor allele frequency; NR: not reported; NT: nucleotide. ^a^ Genomic placement is described using the GRCh37 (hg19) version of the reference genome. ^b^ MAF obtained for Latino/Admixed Americans from gnomAD database v2.1.1 (https://gnomad.broadinstitute.org, accessed on 26 September 2021). ^c^ The functionality prediction of splice site variants was made using the dbNSFP v4.2 in silico prediction algorithm. The functionality prediction of frameshift and in-frame variants was made manually considering the region of the variant. In-frame variants were considered likely deleterious, while frameshift variants were considered as deleterious.

**Table 5 pharmaceutics-14-00944-t005:** Influence of deleterious variants (MAF > 10%) on LDL-c response to statins in FH patients: Multivariate linear regression analysis.

Variant	Allele	*n*	β	SE	*p*-Value
*CYP2C8* rs2071426 g.5932A>G	G allele	92	2.8	3.8	0.456
*CYP3A5*3* rs776746 g.12083G>A	A allele	114	12.9	7.7	0.096
*ABCC1* rs45511401 c.2012G>T	T allele	92	−14.4	6.8	0.038
*SLC22A1* rs72552763 c.1260_1262del	Deletion	92	−1.48	4.1	0.718
*SLCO1B1* rs4149056 c.521T>C	C allele	114	−4.7	4.4	0.288
*SLCO1B3* rs60140950 c.767G>C	C allele	92	−8.2	4.5	0.070

The model was adjusted with the following covariates: body mass index, baseline LDL-c, therapy intensity, and presence of SRAE. *n*: number of patients; β: linear coefficient; SE: standard error; LDL-c: low-density lipoprotein cholesterol; FH: familial hypercholesterolemia; SRAE: statin-related adverse events.

**Table 6 pharmaceutics-14-00944-t006:** Association of deleterious variants (MAF > 1.0%) in PK-related genes with SRAE in FH patients: Multivariate logistic regression analysis.

Variable		No SRAE, % (90)	SRAE, % (24)	OR (95% CI)	*p*-Value
*CYP2C8*4* rs1058930	A allele	45.5 (35)	35.7 (5)	0.70 (0.19–2.37)	0.574
*CYP2C9*2* rs1799853	T allele	16.9 (15)	12.5 (3)	0.54 (0.1–2.2)	0.428
*CYP2C9*9* rs2256871	G allele	2.2 (2)	12.5 (3)	3.03 (0.35–29.74)	0.309
*CYP3A5*6* rs10264272	A allele	3.4 (3)	4.2 (1)	1.34 (0.06–13.48)	0.817
*CYP3A5* rs6977165	G allele	11.2 (10)	12.5 (3)	1.11 (0.22–4.44)	0.886
*CYP3A5*3* rs776746	A allele	93.3 (83)	95.8 (23)	2.7 (0.33–60.01)	0.418
*ABCC1* rs45511401	T allele	6.5 (5)	14.3 (2)	1.65 (0.2–9.46)	0.594
*ABCC2* rs17216317	T allele	5.2 (4)	14.3 (2)	6.12 (0.72–41.6)	0.067
*ABCC2* rs8187692	T allele	5.2 (4)	7.1 (1)	1.28 (0.06–11.08)	0.841
*ABCC3* rs11568591	A allele	13 (10)	14.3 (2)	0.72 (0.07–4.06)	0.734
*SLC22A1* rs35888596	A allele	3.9 (3)	7.1 (1)	3.44 (0.16–32.63)	0.317
*SLC22A1* rs35854239	Deletion	37.7 (29)	14.3 (2)	0.27 (0.04–1.19)	0.122
*SLCO1B1*5* rs4149056	C allele	21.3 (19)	25.0 (6)	1.23 (0.36–3.85)	0.727
*SLCO1B1* rs59502379	C allele	3.4 (3)	4.2 (1)	2.4 (0.11–22.59)	0.479
*SLCO1B3* rs60140950	C allele	26 (20)	14.3 (2)	0.36 (0.05–1.68)	0.252

Each model was adjusted with the following covariates: baseline LDL-c, presence of FH-related variant, and adherence to statin. Number of patients in round brackets. The *p*-value was adjusted using the Benjamini–Hochberg correction. OR: odds ratio; CI: confidence interval; BMI: body mass index; FH: familial hypercholesterolemia; LDL-c: low-density lipoprotein cholesterol; PK: pharmacokinetics; SRAE: statin-related adverse events.

## Data Availability

All relevant data of this study are available within the article and its Appendix A. The raw DNA sequence reads (BioProject accession number PRJNA662090) are available for download at https://www.ncbi.nlm.nih.gov/sra/PRJNA662090, submitted on 8 September 2020. Source data are provided with this paper.

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
