# Peer review of "Genetic Variant ABCC1 rs45511401 Is Associated with Increased Response to Statins in Patients with Familial Hypercholesterolemia"

_pharmaceutics, 2022, doi:10.3390/pharmaceutics14050944_

Round 1

Reviewer 1 Report

This is an interesting and potentially important study of the PGX of statin, lipid lowering drugs to treat faimilial hypercholesterolemia (FH).  The goal was to study PGX variants using exon-targeted gene sequencing in 114 Brazilian patients with FH who had received different doses of statin drugs (atorvastatin, simvistatin, rosuvastatin) at intermediate or high intensity. Responders to treatment were defined as at least 50% decrease in blood lipids while non-responders did not experience 50% decrements in lipids. PGX analyses involved 23 PK genes including transporter and drug metabolizing genes.  Prediction models were used to asses functional impact depending on the type of variant, that included in silico algorithms.  Drug interactions, adherence rate and statin-related adverse events (SRAE) were also included. Numerous variants were associated with FH pathology, functional prediction scores, in silico predictions, SRAE and lipid level responsiveness. From this sea of data, the authors conclude rs45511401 carriers showed improved statin response.  They also provided a molecular basis for this improved responsiveness (Figure 4).  Authors conclude in the abstract (but not in body of paper that i could fine) that this variant could be actionable with regard to individualizing statin treatment.

This is impressive--yet several if not most of the variants having MAFs of < 0.1: in 114 patients getting different statin drugs of variable dosing.  No attempt is made to account for the number of variants tested.  My guess is that if authors applied Bonferoni adjustments they may lose statistical significance.  How do the authors respond to this criticism?

Author Response

Dear reviewer,

We would like to thank you for the discerning review and the valuable contribution that helped to improve our manuscript. We have added the point-by-point responses below. All modifications are highlighted in the revised manuscript.

Thank you for your contribution.

COMMENT: Authors conclude in the abstract (but not in body of paper that i could fine) that this variant could be actionable with regard to individualizing statin treatment.

ANSWER: The text was revised and a sentence regarding main conclusions was added at the discussion (page 19, lines 533-535) and in the conclusions (page 20, lines 581-582).

COMMENT: This is impressive--yet several if not most of the variants having MAFs of < 0.1: in 114 patients getting different statin drugs of variable dosing.  No attempt is made to account for the number of variants tested.  My guess is that if authors applied Bonferoni adjustments they may lose statistical significance.  How do the authors respond to this criticism?

ANSWER: We thank and share reviewer's concern about potential errors due to multiple testing. As a strategy to avoid these misinterpretations, we used Benjamini-Hochberg correction to adjust p-values, considering a false discovery rate (FDR) of 10% (Material and methods, page 5, lines 226-241). The Benjamini–Hochberg method controls the FDR using sequential modified Bonferroni correction for multiple hypothesis testing, considering the total number of tests, but also taking into account the individual p-value’s rank and the FDR [Benjamini & Hochberg; 1994]. Indeed, as Bonferroni only consider the number of tests performed, it "punishes" all input p-values equally, whereas Benjamini-Hochberg "punishes" p-values accordingly to their ranking, avoiding in this way false negatives and false positive results.

Reviewer 2 Report

The authors describe the results of an interesting pharmacogenetic study of statin response in familial hypercholesterolemia (FH) patients. Exome-target sequencing was carried out for a panel of 84 genes already known to be involved in the response to pharmacological treatment. FH but only 23 genes involved in statin pharmacokinetics were analyzed. Annotation and evaluation of the functional impact of deleterious variants on the interaction between the protein and statin were evaluated through 3D modeling and molecular docking. A total of 355 variants were identified, including 41 novel variants. Although no evidence of association with the risk of adverse effects was found for any of the variants evaluated, a deleterious variant at ABCC1 was found to be associated with statin response in Brazilians. Correctly and clearly written manuscript. I only have a few minor comments the authors might want to consider.

Minor comments

  1. Page 2, line 61: The word “focus” is repeated here.
  2. Page 2, line 88: It seems the word “hundred” is repeated.
  3. Exome-target sequencing was performed for a panel of 84 genes known to be involved in FH and related to response to treatment, although the analyses were focused on only 23 genes encoding proteins implicated in the pharmacokinetics of statins. This strategy could help to understand the genetic variants that might be responsible for the modification of the function of proteins already known to be involved in the statin pathway. Nonetheless, it would be very interesting to conduct a hypothesis-free study to attempt the discovery of novel loci (apart from the statin pharmacokinetics) that might be of relevance for the response to this treatment. Whole-exome sequencing could be considered or at least a genome-wide association study using genotype data (and subsequent imputation of genetic variants).
  4. Figure 1: This figure does not seem to be really necessary since it does not provide any additional information from what is already described in the text.

Author Response

Dear reviewer,

We would like to thank you for the discerning review and the valuable contribution that helped to improve our manuscript. We have added the point-by-point responses below. All modifications are highlighted in the revised manuscript.

Thank you for your contribution.

COMMENT: Page 2, line 61: The word “focus” is repeated here.

ANSWER: We appreciate the reviewer’s comment. The text was revised and corrected.

COMMENT: Page 2, line 88: It seems the word “hundred” is repeated.

ANSWER: The text was revised and corrected.

COMMENT: Exome-target sequencing was performed for a panel of 84 genes known to be involved in FH and related to response to treatment, although the analyses were focused on only 23 genes encoding proteins implicated in the pharmacokinetics of statins. This strategy could help to understand the genetic variants that might be responsible for the modification of the function of proteins already known to be involved in the statin pathway. Nonetheless, it would be very interesting to conduct a hypothesis-free study to attempt the discovery of novel loci (apart from the statin pharmacokinetics) that might be of relevance for the response to this treatment. Whole-exome sequencing could be considered or at least a genome-wide association study using genotype data (and subsequent imputation of genetic variants).

ANSWER: We would like to thank the reviewer for the comment. We agree with the reviewer that an exploratory study of the whole genome would be interesting to discover new candidate genes. Our rationale was that genome-wide association studies have been previously performed for statin response (POSTMUS et al, 2014; THOMPSON et al, 2009) and most of the genes shown to be relevant participate in cholesterol homeostasis and statin pharmacokinetics. For statin-related myopathy, a recent meta-analysis of whole-genome sequencing showed that no new variants were associated with this event (FLOYD et al, 2019). For this reason, we aimed to explore the main genes that are mainly related to FH and treatment response and to explore the interaction between them. This study is also part of a bigger project that includes sequencing of a panel of 84 genes associated with the pathophysiology of familial hypercholesterolemia and statin pharmacokinetics and pharmacodynamics (BORGES et al, 2021). This is indeed an important topic and we have added a comment on the discussion session about this approach (page 20, lines 569-576).

COMMENT: Figure 1: This figure does not seem to be really necessary since it does not provide any additional information from what is already described in the text.

ANSWER: We agree with the reviewer and we removed Figure 1.

Round 2

Reviewer 1 Report

Authors appropriately responded to my comments